# Poplar Sawdust Stack Self-Heating Properties and Variations of Internal Microbial Communities

**DOI:** 10.3390/ma15031114

**Published:** 2022-01-31

**Authors:** Zitao Yuan, Wenbin Xu, Zili He, Hao Shen

**Affiliations:** 1Guangdong Key Laboratory of Environmental Catalysis and Health Risk Control & Guangzhou Key Laboratory Environmental Catalysis and Pollution Control, School of Environmental Science and Engineering, Institute of Environmental Health and Pollution Control, Guangdong University of Technology, Guangzhou 510006, China; xuwenbin@gdut.edu.cn (W.X.); zilihe@hotmail.com (Z.H.); 2Guangdong Provincial Key Laboratory of Fire Science and Technology, School of Intelligent Systems Engineering, Sun Yat-sen University, Guangzhou 510006, China

**Keywords:** sawdust stack, Critical Ambient Temperature (CAT), self-heating process, physicochemical properties, microbial communities

## Abstract

The heat accumulation generated by microbial metabolic activities during the storage of the sawdust may lead to spontaneous combustion accidents. This paper studied the Critical Ambient Temperature (CAT) variation of poplar sawdust at different stack dimensions and investigated the physicochemical properties as well as microbial community dynamics during the self-heating process of poplar sawdust stacks. From the self-heating substances test experiments and Frank-Kamenetskii (FK) theory, it was found that the CAT of poplar sawdust stacks would decrease from 158.27 °C to 102.46 °C with the increase of stack size from 0.1 m to 3.2 m. From the sawdust stack self-heating experiments, microbial metabolic activities were enhanced with the increasing moisture content (by watering) and oxygen (by turning over), which led to a remarkable increase of the sawdust stack temperature and the rapid decomposition of biochemical components (especially cellulose and hemicellulose). From the microbiological community analysis, at the thermophilic stage (around 60 °C, large amounts of heat release in compost bin), the existence of thermostable bacteria (such as *Brevibacillus thermoruber*, *Bacillus thermoamylovorans* and *Paenibacillus barengoltzii* belonging to *Firmicutes*) played an important role in degrading organic substances. The heat generated by the microbial metabolic activities might lead to spontaneous combustion eventually if sawdust stack is large enough. Therefore, the sawdust should be stacked in a cool and dry area while avoiding large amounts of storage in high humidity environments.

## 1. Introduction

Large quantities of lignocellulosic wastes are generated by the logging industry and agriculture [1]. These lignocellulosic wastes, including sawdust, straw, bagasse and so on, are valuable for energy utilization and soil nutrient replenishment, and are used as feedstock for biofuels and growth substrates for mycelium-based composites [2,3]. However, since sawdust and some other lignocellulosic wastes are usually stored in the form of large piles in the outdoor environment, it may lead to spontaneous combustion and harm the environment if there is no proper management [4,5].

Previously, Hogland and Marques [6] revealed that lignocellulosic waste displays a self-heating phenomenon, in which biological processes played a dominant role when the temperature was between 20 °C and 80 °C. Jurado et al. [7] showed that in composting ecosystems, the aerobic microbial metabolism could make temperature increase over 50 °C during the composting process, and then maintain a high temperature with fluctuations until most of the biodegradable nutrients were depleted. The degradation of lignocellulosic wastes is also attributed to the metabolism of indigenous microorganisms. Xu et al. [8] had found that when the dimensions of a stack of sawdust increased from 0.1 to 1.6 m, the Critical Ambient Temperature (CAT) of *Dalbergia cochinchinensis*, *Cupressus funebris*, *Larix olgensis* all would gradually decrease, especially for *Dalbergia cochinchinensis*, whose CAT could reduce from 155 °C to 81.2 °C. It can be found from the above studies that the heat accumulated by microbial metabolism might reach the CATs of sawdust stacks, which would then cause thermal decomposition and even spontaneous combustion. Therefore, it is necessary to obtain the characteristics of the spontaneous combustion of sawdust stack. It is also necessary to further study under what storage conditions sawdust stacks are more prone to self-heating as well as what physical and chemical changes occur during the self-heating process.

In the field of microbial community dynamics of lignocellulosic waste stacks, different microbial populations dominate at various stages of composting and have distinct roles in the degradation of organic matter [9,10]. Some researchers further studied the diversity and successions of bacteria and fungus during the process of co-composting of organic wastes [11,12,13]. A previous study determined that the abundance and structure of the bacterial community had distinctly different responses to different nutritional treatments [14]. Moreover, Zhao et al. [15] and López-González et al. [16] investigated the evolution of organic matter and its association with microbial community dynamics during lignocellulosic waste composting. They found that both bacteria and fungus communities significantly affected the abundance of oxygen-containing functional groups and humic-like materials. Jurado et al. [7] described an attempt to identify and select specific microorganisms involved in the composting of lignocellulosic materials and exploit the isolated microorganisms for diverse uses such as enzyme production, compost inoculants and composting bioaugmentation. These papers mainly studied the microbial community dynamics of composting, which aimed at achieving lignocellulosic waste recycling and make composting more efficient. However, despite broad research in this field of microbial community dynamics of lignocellulosic wastes, there is still lack of information on the relationship between lignocellulosic wastes stack self-heating and microbial community dynamics.

The aims of this study were: (i) to investigate the variation of the Critical Ambient Temperature of sawdust storage stacks of different sizes; (ii) to reveal the variation of physicochemical properties during sawdust stack self-heating process by analyzing the changes of moisture content, pH and biochemical components. (iii) to study the evolution and metabolic characteristics of the microbial communities during the self-heating process of sawdust stacks.

## 2. Experiments

### 2.1. Materials

In this paper, poplar sawdust (*Populus *L.; Guangxi, China) with particle size less than 0.25 mm was used as the sawdust material of all experiments. The moisture content of the raw sawdust was about 8.00%. The cellulose, hemicellulose and lignin of raw poplar sawdust measured by the Van Soest method [17,18] were 45.92%, 20.89% and 25.44%, respectively. The autoignition temperature of poplar sawdust is 238.5 °C, which was tested by solid spontaneous ignition point tester (model: HY 21756, Jilin Hongyuan Scientific Instrument Co. Ltd., Jilin, China) that was designed according to the Chinese standard (GB/T 21756-2008) [19] and Normes Francaises (NF T20-036) [20].

### 2.2. Self-Heating Substances Test Experiments

The self-heating experiments of poplar were implemented with the self-heating substances tester (model: HWP01-10E, Hangzhou Young Instrument Science & Technology Co., Ltd., Hangzhou, China). The instrument provides two cube-shaped sample baskets (25 mm and 100 mm) for the self-heating experiments and uses two thermocouples to record the temperature variation of the oven and the center of the sample basket respectively. If the sample temperature was 60 °C higher than the oven temperature, the system would indicate that thermal decomposition had occurred. Based on this principle, it is possible to judge whether the sample is a self-heating substance according to the classification of self-heating substances [21]. The experiments started at 140 °C and then were carried out at different ambient temperatures with intervals of 10 °C (such as 150 °C, 160 °C and 170 °C) in order to obtain the CAT of the sawdust samples in the 100 mm cube basket.

### 2.3. Thermogravimetry (TG) Analysis

In this study, the TG curve of sawdust samples were obtained by the simultaneous thermal analyzer (model: TG/DSC 3+, Mettler Toledo, Zurich, Switzerland). There were four equal parts of samples (about 5 mg each) in the experiments, which were all placed in the air atmosphere with a flow rate of 50 mL/min and used the open alumina crucible as sample pan. The four samples were treated at the heating rate of 2, 5, 10, 20 °C/min, respectively, and the weight changes in a temperature range of 30–400 °C were observed.

### 2.4. Sawdust Stack Self-Heating Experiments

In this study, a self-heating experiment was carried out to study the self-heating process of sawdust stacks. As the moisture content of the sawdust sample is too low for the self-heating experiment, water was added to make the sawdust sample wet enough. Thus, the moisture content of sawdust stack is reach about 53.49% before the sawdust stack self-heating experiment.

The schematic diagram of sawdust stack self-heating experiment is shown in Figure 1. About 60 kg wet sawdust sample was placed in the 0.78 m length × 0.60 m width × 0.80 m height compost bin. There are many small air holes on the four sides of the compost bin, which provided air for the sawdust stack self-heating process. Two temperature monitoring points were selected as follow: surface (0.05 m below sample surface) and inside (0.4 m below sample surface). Sample temperatures were recorded by the data acquisition, and the ambient temperatures were monitored with a thermometer.

The sawdust stack self-heating experiment was done continuously for 51 days. During the 51 days, the ambient temperature varied from 21.5 °C to 33.4 °C and the mean temperature was 27.0 °C, the relative humidity of the ambient air ranged from 25.7% to 84.0% and the average humidity was 61.5%.

The 51-day sawdust stack self-heating experiment could be divided into three phases, named A, B and C, respectively. As shown in Table 1, at the beginning of each phase, the sawdust samples in the compost bin were treated (at day 0, day 17 and day 34) and then stacked continuously for 17 days, respectively. The details of treatments were as follows: (1) at the beginning of phase A, the initial moisture content of the sawdust stacks adjusted to about 53.49% by watering and the sawdust stack turned over to place the sawdust in full contact with the air; (2) at the beginning of phase B, only adequately turned over the sawdust stack in the compost bin (no watering); (3) at the beginning of phase C, the initial moisture content of sawdust stack was adjusted to 53.00–55.00% by watering and the sawdust stack turned over again.

Some sawdust samples were collected according to temperature changes throughout the self-heating experiment. There are seven critical days which were chosen for sampling: (1) day 0 (the beginning stage), (2) day 3, day 20 and day 37 (the temperature rose quickly), and (3) day17, day 34 and day 51 (the temperature remained stable). The sampling positions were the same as that of the temperature monitoring points. The sampling days and sample ID are shown in Table 1. Each sample was collected in quadruplicate from the inside or surface. One of the samples weighed about 15 g and was stored at 4 °C for subsequent physicochemical analysis, and the other three were all about 10 g and preserved at −80 °C for DNA extraction.

### 2.5. Physicochemical Analysis

The physicochemical analysis of the collected samples in this paper included the determination of moisture content, pH and biochemical components such as cellulose, hemicellulose and lignin. For the pH measurement, 45 mL of distilled water was added to 5 g of collected sample and agitated for 5 min and then the pH of the solution was determined by a pH analysis pen (model: ST20, OHAUS, Parsippany, NJ, USA) [22]. The moisture content of collected samples was determined by measuring the weight loss after oven-drying at 105 °C for 24 h.

Previously, Biswas et al. [23] had found that the main chemical components of poplar sawdust were glucan, xylan and lignin, among which glucan and xylan were representative polysaccharides in hemicellulose. In this paper, the contents of biochemical components in the collected sawdust samples were measured by the Van Soest method [17,18]. Hemicellulose content was calculated from the difference between Neutral Detergent Fiber (NDF) and Acid Detergent Fiber (ADF). Cellulose content was calculated from the difference between ADF and Acid Detergent Lignin (ADL). Lignin content was calculated from the difference between ADL and ash content.

### 2.6. Microbiological Community Analysis

#### 2.6.1. DNA Extraction, Amplification, and Sequencing

Each collected sample was in triplicate, and they were used to extract genomic DNA by the NucleoSpin Soil kit (Machery-Nagel, Düren, Germany). Subsequently, DNA concentrations and purity were quantified by a Qubit Fluorometer (model: Qubit4.0, Thermo Fisher, Waltham, MA, USA).

In order to determine the diversity and composition of the bacterial communities in each collected sample, the V4 hypervariable region of the 16S rRNA gene was amplified from the genomic DNA using universal primers: 515F (5′-GTGCCAGCMGCCGCGGTAA-3′) and 806R (5′-GGACTACHVGGGTWTCTAAT-3′). Fungus communities were characterized by amplifying ITS genes, which were amplified using primers its1 (5′-CTTGGTCATTTAGAGGAAGTAA-3′) and its2 (5′-GCTGCGTTCTTCATCGATGC-3′). The amplification was carried out using Phusion High-Fidelity PCR Master Mix (New England Biolabs, Hitchin, UK). All PCR reaction mixtures contained 4 μL each primer, 25 μL PCR master mix, 30 ng of template DNA, and sterile water up to a final volume of 50 μL. The thermal cycling of PCR program was as follow: initial denaturation at 98 °C for 3 min, followed by 30 cycles of 98 °C for 45 s, 55 °C for 45 s, and 72 °C for 45 s; and last, 72 °C for 7 min. The PCR products were purified with Agencourt AMPure XP beads (Beckman Coulter, Brea, CA, USA) to remove the unspecific products and the clean DNA was sequenced with the Illumina Hiseq system (model: Hiseq 2500 PE250 platform, Illumina, San Diego, CA, USA) at BGI Genomics Co., Ltd. (Wuhan, China).

#### 2.6.2. Processing of Sequencing Data

Raw data generated from the HiSeq 2500 system were assembled using Fast Length Adjustment of Short reads (FLASH, Version 1.2.11). In order to control the quality process, raw tags were then filtered to obtain high-quality clean tags using the Quantitative Insights in the Microbial Ecology (QIIME) software package (Version 1.7.0) [24,25]. In order to detect and remove chimeric sequences, the tags were compared to the Gold database (16S rRNA chimeric database) and UNITE (ITS chimeric database) using the UCHIME algorithm (Version 4.2.40) [26]. Then, a set of effective tags were obtained. Operational taxonomic units (OTUs) (97% similarity) sequences were aligned by using UPARSE with these effective tags. The Greengene Database (16S rRNA) and UNITE Database (Version 7.2) (ITS) were used to annotate the taxonomic information for each OTU representative sequence by using the RDP classifier (Version 2.2) [27]. OTUs abundance information was normalized with a standard sequence number whose sample contained the least number of sequences [28]. Chao richness estimate was calculated based on the observed species and analyzed in QIIME. Heatmaps, which were used to show the relative abundance of bacterial and fungal communities, were generated from the relative abundance of OTUs by using Gplots Package in R 3.1.1.

## 3. Results and Discussion

### 3.1. Critical Ambient Temperature (CAT)

#### 3.1.1. CAT of 100 mm Cube Sawdust Sample

In the self-heating substances test experiments, the temperature variation of poplar sawdust stack in the 100 mm cube basket under Lower Critical Ambient Temperature (LCAT) and Upper Critical Ambient Temperature (UCAT) are shown in Figure 2. In the figure, the LCAT refers to the oven temperature at which the sample temperature would remain stable and the final temperature is not 60 °C higher than the oven temperature. On the other hand, at the UCAT, the sample temperature would rise rapidly and be over 60 °C than the preset oven temperature at a certain moment. At the UCAT, the samples would experience a process transferred from smoldering to burning and remained only a few ashes finally.

The mean temperature of LCAT and UCAT was regarded as the CAT of sawdust samples [8]. From Figure 2, the LCAT, UCAT and CAT of poplar sawdust samples obtained by the self-heating substances test experiments were 170 °C, 180 °C and 175 °C. It was found that the dangerous self-heating reaction did not take place at 140 °C, which could demonstrate the poplar sawdust is not a self-heating substance according to the classification of self-heating substances [21]. However, for the sawdust stack samples in 100 mm cube basket, the self-heating reaction occurred among 170 °C and 180 °C, which reminds that the spontaneous combustion risk of the sawdust stack still existed and should not be ignored for large sawdust stacks. The CATs of poplar sawdust under larger stack dimensions were still unknown, and will be discussed in the next section.

#### 3.1.2. CAT of Larger Sawdust Stack Derived by Frank-Kamenetskii (FK) Theory

The FK theory can reflect the temperature distribution within a stack of substances [29]. The CAT of sawdust stacks can be calculated according to the FK theory. In the CAT calculation, activation energy should be calculated firstly. The activation energy is an important factor in reaction kinetics, which can be regarded as the energy barrier that the reaction needs to overcome [30]. The activation energy of a sawdust sample can be calculated by the kinetics based on the TG analysis. For a solid reaction, the non-isothermal kinetic equation for solid reaction can be expressed as follows:(1)$$\frac{d\alpha }{dt}=A\cdot \exp\left(-\frac{E}{RT}\right)\cdot f\left(\alpha \right)$$

In the above equation, α refers to the degree of reaction, %, which can be defined as the mass fraction of the decomposed solid; $f\left(\alpha \right)$ is a reaction model that represents a certain solid state mechanism, it would be different in different reactions [31]; *R* is the universal gas constant, 8.314 J·mol^−1^·K^−1^; A is the pre-exponential factor; $\frac{d\alpha }{dt}$ is the reaction rate that is the first derivative of the degree of reaction with respect to time; *T* is the ambient temperature, K; *E* is the reaction activation energy of sample, kJ/mol.

In this study, the Friedman method was used to calculate the activation energy *E*, which was recommended by the International Federation of Thermal Analysis and Calorimetry (ICTAC) [32,33]. The Friedman method was very suitable for the activation energy calculation as it could avoid the influence of the model function [34,35]. The Friedman method takes the natural logarithm of both sides of Equation (1), as shown in Equation (2):(2)$$\ln\left(\beta \frac{d\alpha }{dT}\right)=\ln\left(A\cdot f\left(\alpha \right)\right)-\frac{E}{RT}$$

In the above equation, β is the heating rate of TG, K·min^−1^ and the TG curves of samples were shown in Figure 3. The mean value of *E* at different reaction degrees was regarded as the activation energy of this substance. As neither the initial stage nor the completion stage of the reaction can represent the entire thermal decomposition process [36,37], it is not necessary to pay too much attention to the kinetic at the higher reaction degree (over 50%) when studying the problem of spontaneous combustion. Therefore, the reaction degrees of 20%, 30%, 40% and 50% were selected in this study. The least square method is used to perform linear fitting to obtain the linear slope value which is the activation energy *E*. The activation energies of poplar sawdust samples calculated by Friedman method are shown in the Table 2. The activation energy of poplar sawdust samples calculated by the Friedman method was 140.53 (±3.7) kJ/mol.

From the FK theory, Equation (3) provided the definition formula of critical Frank-Kamenetskii parameter $\delta_{cr}$, whose value was studied previously [29,38] and only related to the stacked shape. Here it was assumed that all samples were stacked in a cube shape, and the value of $\delta_{cr}$ was taken as 2.52:(3)$$\delta_{cr}=\frac{Q\cdot E\cdot l^{2}\cdot f\left(\alpha \right)}{\lambda \cdot R\cdot T_{a,cr}^{2}}\cdot \exp\left(\frac{E}{R\cdot T_{a,cr}}\right)$$

For spontaneous combustion problems, the effect of reaction model $f\left(\alpha \right)$ could be ignored, because the initial stage of spontaneous combustion does not consume much fuel. The universal gas constant *R*, thermal conductivity λ and total heat Q are all constants. Therefore, the CATs of sawdust samples just be affected by the stack dimension l and the activation energy *E*, as shown in Equation (4). As the activation energy *E* of the sawdust sample has been calculated and the CAT of the sawdust sample in the 100 mm cube basket have been obtained with experiments, the CAT values of poplar sawdust under different stack dimensions were derived by Equation (4), as shown in Table 3:(4)$$l^{2}\propto T_{a,cr}^{2}\cdot \exp\left(\frac{E}{R\cdot T_{a,cr}}\right)$$

From the CATs of sawdust stacks of different dimensions (from 0.1 m to 3.2 m), it can be found that the CAT value of the poplar sawdust will decrease as the stack dimension increases. When the stack dimension of poplar sawdust is 3.2 m, the CAT value is only 102.46 °C.

### 3.2. Physicochemical Properties

#### 3.2.1. Temperature Evolution of Sawdust Stack

Temperature is the most important parameter used to monitor the self-heating risk of sawdust stacks. As shown in Figure 4, the self-heating experiments of sawdust stacks could be divided into three phases (A, B and C), and it could be found that there were some significant differences in the temperature evolution for these three phases.

For phase A, both the inside and surface sawdust temperatures rose simultaneously after watering and turning over on day 0 and then reached their maximum values on day 3, which showed the inside sawdust temperature increased from 30.7 °C to 62.0 °C and the surface sawdust temperature increased from 30.5 °C to 58.5 °C. This stage was called the thermophilic stage, and may be not only attributed to the abundant and active indigenous microorganisms in the raw sawdust stack, but also attributed to the availability of sufficient nutrients (such as oxygen, water and so on) needed by indigenous microorganisms for metabolic activities at the beginning of the experiment. After that, both the inside and surface sawdust temperatures decreased quickly between days 4 and 9, which showed the inside sawdust temperature decreased from 62.0 °C to 32.8 °C while the surface sawdust temperature decreased from 58.5 °C to 31.8 °C. From day 10 on, both the inside and surface sawdust temperatures slowly returned to ambient temperature and remained stable until the end of phase A.

For phase B, both the inside and surface sawdust temperatures increased slightly (about 4.4 °C and 2.8 °C, respectively) from days 17 to 20 after adequately turning over the sawdust in the compost bin on day 17. This phenomenon might be explained by the fact that turning over the sawdust stack could effectively increase the oxygen content required for microbial metabolic activities in the compost bin.

For phase C, both the inside and surface sawdust temperatures increased significantly from days 34 to 37 after watering and turning over on day 34, and the inside sawdust temperature increased from 30.5 °C to 41.5 °C while the surface sawdust temperature increased from 27.8 °C to 33.5 °C. The temperature increase range of the inside and surface sawdust from days 34 to 37 of phase C was between that of from days 0 to 3 of phase A and from days 17 to 20 of phase B.

During the whole sawdust stack self-heating process, it could be found that the evolution of sawdust stack temperatures would fluctuate approximately with the ambient temperature for most of the time. When the ambient temperature was relatively high, it was easier to cause the high temperature of inside sawdust stack, which would lead to an increase of fire accident possibility risk of the sawdust stack.

#### 3.2.2. Changes of Moisture Content and pH

The moisture content variation of sawdust samples during the self-heating experiment are shown in Figure 5a. After watering and turning over on day 0 of phase A and on day 34 of phase C, the initial moisture content of the sawdust stacks in the compost bin were both adjusted to 53.00–55.00%, and the temperature of the sawdust stack increased to around 60 °C and 40 °C quickly within 3 days. However, although the sawdust stack of phase B was also turned over on day 17, it was not watered and its initial moisture content was only about 45.25%, the temperature rise in this phase was not so obvious as that in other two phases. This indicated that higher initial moisture content would lead to greater self-heating of sawdust stacks.

The pH variation of sawdust samples during the self-heating experiment is shown in Figure 5b. Previously, Amir et al. [39] revealed that the optimum pH range for the growth of microbial populations was from 5.50 to 8.00. Similar results were observed in the present study, where the pH values ranged from 5.47 to 6.19 through the whole sawdust stack self-heating process. The change trends of pH value were similar in the three phases. The pH values firstly increased within 3 days, and then decreased slowly until the end of each phase. On the one hand, the pH rise might be caused by the degradation of organic nitrogen [40] during the initial self-heating process. On the other hand, the pH drop could be attributed to the bioconversion of the organic material into other intermediate species like organic acids [41]. In a word, the results of pH variation revealed that the increasing of temperature would be accompanied by a slight increase of pH, and when the temperature decreased or remained stable, the pH would decrease slowly.

#### 3.2.3. Decomposition of Cellulose, Hemicellulose and Lignin

The self-heating of sawdust stacks would be closely related to the microbial decomposition of the main components, namely cellulose, hemicellulose and lignin. Table 4 showed the component content variations of the inside and surface sawdust samples collected from the compost bin during the self-heating process. It was found that the content of the three main components, including cellulose, hemicellulose and lignin were all decreased while the others gradually increased. Among the three main components, the decomposition rate of lignin was much lower than that of cellulose and hemicellulose during the whole sawdust stack self-heating process. This might be due to the recalcitrant nature of lignin, which acted as an integral cell wall constituent [42].

In order to analyze in-depth the relationship between the temperature variation and the sawdust main component decomposition, the daily average decomposition rates of cellulose, hemicellulose and lignin of the inside and surface sawdust samples during the self-heating process are listed in Table 5. During the first three days of each phase (days 1–3, days 18–20 and days 35–37), when the inside and surface sawdust sample temperatures were increasing, higher daily average decomposition rates were observed for cellulose, hemicellulose and lignin. When the inside and surface sawdust sample temperatures were decreasing or remained stable (days 4–17, days 21–34 and days 38–51), the daily average decomposition rates in cellulose, hemicellulose and lignin were much lower. The reason may be that most easily degradable organic matter had been metabolized, and the water and oxygen content in the compost bin were consumed a lot in the former of each phase. On the other hand, among the first three days of each phase, the largest temperature rises as well as the largest daily average decomposition rates of cellulose, hemicellulose and lignin arose simultaneously at days 1–3 of phase A. Obviously, as there are better microbial activity conditions including moisture content, oxygen content and fresh sawdust nutrition, the metabolic activity of microorganisms in phase A was much stronger and more growth substrates were consumed.

### 3.3. Microbial Communities

#### 3.3.1. Diversity of the Microbial Communities

In order to reflect the relationship between temperature variation and microbial community evolution, the Illumina high-throughput sequencing method was used to analyze the microbial communities of sawdust samples (Table 1) collected from the different stages of the sawdust stack self-heating process. The changes of the bacteria and fungus communities in 13 sawdust samples (each in triplicate) were analyzed based on a total of 1,281,841 and 1,196,110 sequences with an average of 32,868 (30,029–36,633) and 30,669 (30,057–31,048) per sample, respectively. These sequences were clustered into 724 and 162 OTUs by the average neighbour algorithm with a cut-off of 97% similarity for bacteria and fungus communities, respectively.

The OTU numbers of bacteria and fungus communities in all collected samples are shown in Figure 6a,b, respectively. Obviously, in the same sample, the OTU number of bacterial communities was higher than that of the fungus communities. Moreover, the OTU numbers of bacteria and fungus communities in raw material (CK) were relatively higher, which proved that the indigenous microorganism community in raw material was very rich. On day 3 of phase A when the surface and inside sawdust sample temperatures approached or exceeded 60 °C, it was found that the OTU numbers of bacteria and fungus communities were both decreased dramatically. The reason might be that it was difficult for many microorganisms to survive at such high temperatures. After turning over at the beginning of phase B, the OTU numbers of bacteria and fungus communities were all increased at day 20 and then remained stable until the end of phase B. This might be related to the relatively stable temperature of the phase B and the suitable growth of microorganisms at 23.7 °C to 33.8 °C. During the phase C, the re-supply of water and oxygen made the OTU numbers of bacteria and fungus communities increase continuously. It could be seen that the temperature at this phase was between 25 °C and 42 °C, which was also suitable for the growth of microorganisms.

The diversity indices, namely Good’s coverage and Chao richness were calculated based on the OTUs of each sample. There was no significant difference in calculated sampling coverage among the three phases. The coverage values of all samples were no lower than 0.99, which means that the sequencing depth was sufficient to cover most of the microorganisms. The Chao index could reflect the species richness of samples. The larger the Chao index value, the richer the species in the sample. As shown in Figure 6c,d, a significant difference between the bacterial and fungal communities among samples was also found in the Chao richness estimator, and the result of which was consistent with the above analysis of the OTU number.

#### 3.3.2. Changes in Microbial Community Composition

The normalized sequences of 16S rRNA of bacteria and ITS of fungus were aligned against the Greengene database and UNITE database, respectively, and clustered into phylum taxonomic level. Of the classifiable sequences, 22 phyla, 17 classes, 28 orders, 45 families, 46 genera and 26 species were identified for bacteria, while three phyla, seven classes, 13 orders, 17 families, 21 genera and 22 species of fungi were identified. As for the classified sequences, their phylum and species levels were analyzed, which could well reveal the relationship between temperature variation and microbial community composition.

Figure 7a showed the bacteria community composition at the phylum level. *Proteobacteria*, *Firmicutes*, *Bacteroidetes*, *Actinobacteria*, *Verrucomicrobia* and *Chloroflexi* were the dominant bacterial communities, which accounted for at least 99.995% of the total bacteria communities. The results were in accordance with the data presented previously [11,43]. In the CK sample, the dominant bacteria phyla included *Firmicutes* (51.66%), *Bacteroidetes* (27.96%) and *Proteobacteria* (18.37%). The bacterial community composition changed significantly on day 3 in phase A when the temperatures of inside and surface sawdust stack rose sharply to 62.0 °C and 58.5 °C, respectively. The *Firmicutes* in the I03 increased from 51.66% to 98.13%, while the *Proteobacteria* in the I03 decreased from 18.37% to 1.86%. On the contrary, the *Proteobacteria* in the S03 were higher up to 99.25%, while the *Firmicutes* in the S03 were only 0.71%. The *Bacteroidetes* in both I03 and S03 decreased sharply to below 0.03%. *Firmicutes* is known to grow at high temperatures and widely distributed especially in the thermophilic stage of stacking of agricultural biomass since it could produce spores to resist the extreme conditions [44,45]. As shown in Figure A1, within the phylum *Firmicutes*, three species with particularly high abundance were detected in I03 (thermophilic stage), namely, *Brevibacillus thermoruber*, *Bacillus thermoamylovorans* and *Paenibacillus barengoltzii*, which were commonly found in mesophilic and thermophilic stages [7,46,47]. These species contained thermostable organic substances-degrading enzymes which could quickly degrade organic substances like biochemical components, leading to a large accumulation of heat in the compost bin [48]. Therefore, when the water and oxygen were sufficient (treatment at day 0), the metabolic activities of these bacteria increased, and the temperature of I03 increased quickly as organic substances were degraded, which also indicated that heat accumulation was more likely to occur inside the compost bin. As the sawdust stack temperature in the compost bin gradually decreased, the bacterial community composition at day 17 (I17 and S17) also changed. For the phase B, the bacterial diversity of samples at day 20 (I20 and S20) increased after treatment at day 17. The bacterial composition of samples on day 20 and day 34 was similar and had a relatively stable proportion of *Proteobacteria* (62.64–69.24%), *Bacteroidetes* (12.97–27.18%), *Firmicutes* (6.65–17.25%), *Actinobacteria* (1.70–3.38%), *Verrucomicrobia* (0.06–0.13%) and *Chloroflexi* (0–0.06%). These results reflected that phase B was a relatively stable phase which was consistent with the results of the temperature variation and alpha diversity of the microbial community. During the phase C, the bacteria diversity of samples further increased after treatment on day 34. The *Proteobacteria* was the most abundant (37.02–54.38%) in all samples of the phase C. Following the *Proteobacteria*, the other two dominant phyla were *Bacteroidetes* (23.59–41.38%) and *Firmicutes* (6.01–14.01%). Compared to the phase B, the proportion of *Actinobacteria* (1.64–4.91%), *Verrucomicrobia* (0.90–8.91%) and *Chloroflexi* (0.17–4.58%) in phase C was significantly increased. These results showed that the temperature between 25 °C and 42 °C was suitable for the growth of most bacteria during the sawdust stack self-heating process, especially for the species of *Stenotrophomonas maltophilia* belong to phylum *Proteobacteria*, as shown in Figure A1, which contained organic substances-degrading enzymes and could degrade organic substances [49].

Figure 7b showed the fungal community composition at the phylum level. The dominant communities, *Ascomycota* and *Basidiomycota*, accounted for at least 99.995% of the total fungal communities. The results are in accordance with the previously presented data [50]. In the CK sample, the relative abundance of *Ascomycota* was higher up to 98.98%. After treatment at day 0 of phase A, the fungus composition proportion of I03 almost remained unchanged. However, the proportion of *Basidiomycota* in S03 increased from 1.02% to 19.63% while the proportion of *Ascomycota* in S03 decreased from 98.98% to 80.37%. During phase B, the proportion of *Ascomycota* and *Basidiomycota* remained stable. After treatment at day 34 of phase C, there was a significant difference in the fungal composition proportion between I37 and S37. The *Basidiomycota* sharply increased and became the most dominant fungal phylum in I37 (58.11%) while the *Ascomycota* still was the most dominant fungal phylum in S37 (90.40%). As shown in Figure A2, six species with high relative abundance which contained organic substances-degrading enzymes were detected throughout the sawdust stack process, namely, *Petriella setifera*, *Aspergillus rugulosus*, *Trichoderma longibrachiatum*, *Aspergillus tamarii*, *Aspergillus flavus* and *Aspergillus proliferans* [7,50,51]. They all belong to the phylum *Ascomycota*.

Consequently, during the sawdust stack self-heating process, the sawdust stack temperature between 25 °C and 42 °C was suitable for the growth of most microorganisms, especially for *Stenotrophomonas maltophilia* belonging to phylum *Proteobacteria* and several fungi belonging to the phylum *Ascomycota*. Even when the sawdust stack temperature reached around 60 °C, abundant thermostable bacteria still existed, especially *Brevibacillus thermoruber*, *Bacillus thermoamylovorans* and *Paenibacillus barengoltzii* belonging to the phylum *Firmicutes*. These different microbial populations dominated at different stages of the sawdust stack and had distinct roles in the degradation of organic substances.

## 4. Conclusions

(1)From the self-heating substances test experiments and Frank-Kamenetskii (FK) theory, for stacks of poplar sawdust, with the stack dimensions increased from 0.1 m to 3.2 m, the CAT of the sawdust stack would decrease from 158.27 °C to 102.46 °C. Apparently, the dimensions of sawdust stacks have a significant effect on the spontaneous combustion risk of sawdust stacks.(2)From the sawdust stack self-heating experiments, increasing water and oxygen would enhance microbial metabolic activities, which leads to a striking increase of the sawdust stack temperature and the quicker decomposition of the biochemical components (especially cellulose and hemicellulose) within three days after treatment.(3)From the microbial community analysis, different microbial populations dominated in the sawdust stacks at different stages. During the thermophilic stage (around 60 °C, large amounts of heat release in the compost bin), the dominant thermostable microbes (especially *Brevibacillus thermoruber*, *Bacillus thermoamylovorans* and *Paenibacillus barengoltzii* belonging to *Firmicutes*) played an important role in efficiently degrading organic substances.

In summary, if a sawdust stack is large enough and under appropriate microbial activity conditions (such as moisture content, oxygen content and fresh nutrition), the CAT of the poplar sawdust stack would likely drop below 100 °C or even lower, and the metabolic activities of heat-resistant microorganisms would remain active at that temperature. The heat generated by the microbial metabolic activities might lead to spontaneous combustion eventually if the sawdust stack is large enough. Therefore, the sawdust should be stacked in a cool and dry area while avoiding storage of large amounts in high humidity environments.

## Figures and Tables

**Figure 1 materials-15-01114-f001:**
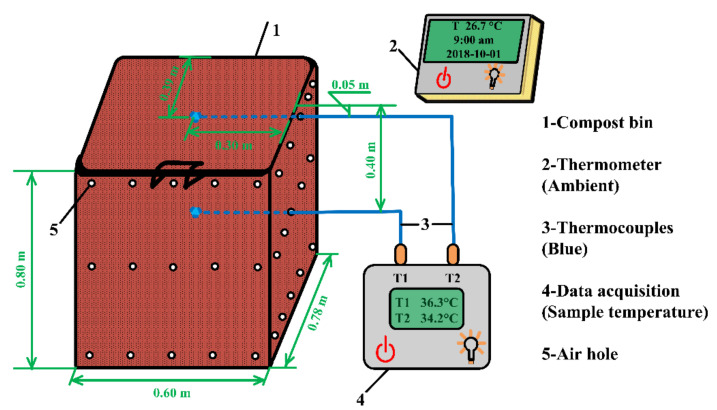
Schematic diagram of sawdust stack self-heating experiment.

**Figure 2 materials-15-01114-f002:**
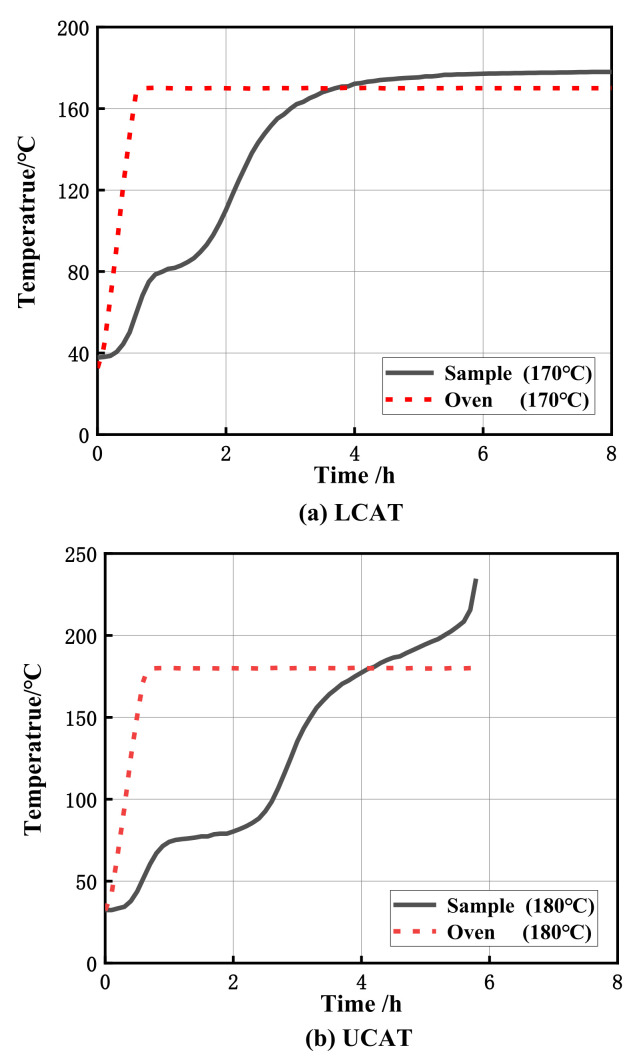
The variation of the oven temperature and the sample temperature (in the center) at the (**a**) LCAT and (**b**) UCAT.

**Figure 3 materials-15-01114-f003:**
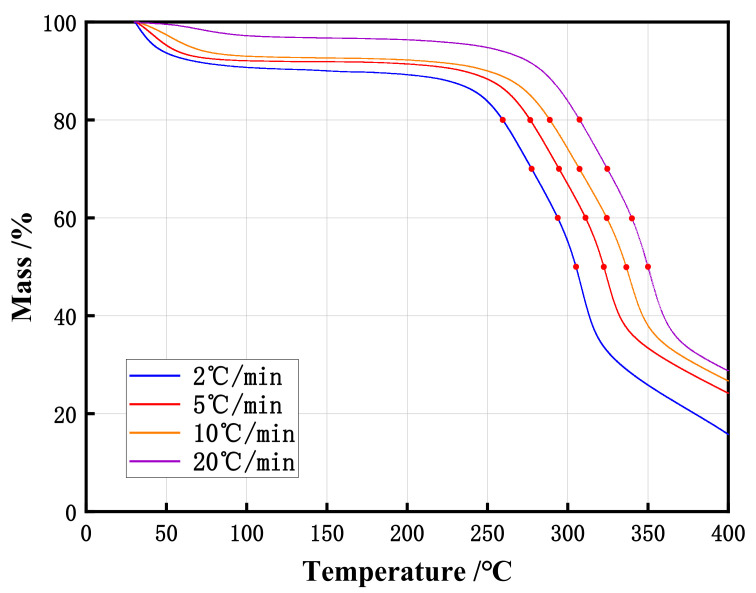
The TG curves of poplar sawdust samples at different heating rates.

**Figure 4 materials-15-01114-f004:**
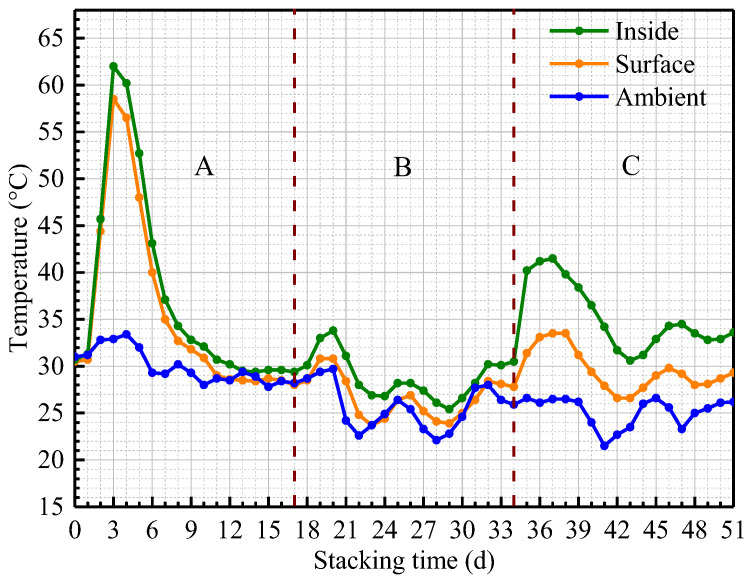
Temperature evolution of sawdust stack self-heating experiment.

**Figure 5 materials-15-01114-f005:**
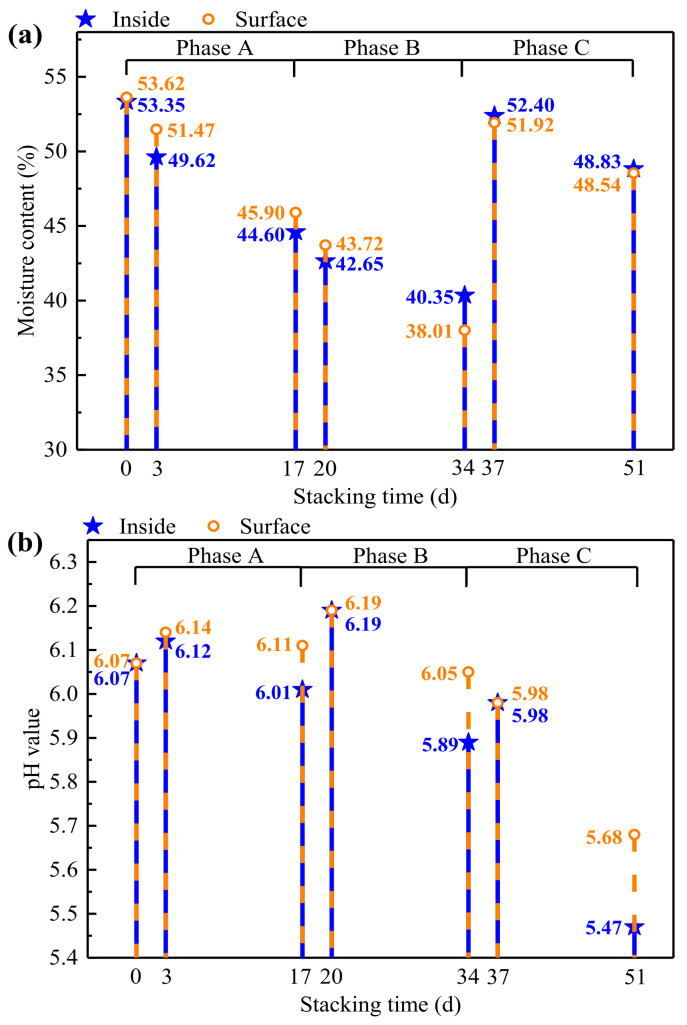
The (**a**) moisture content variation and (**b**) pH value variation during the sawdust stack self-heating experiment.

**Figure 6 materials-15-01114-f006:**
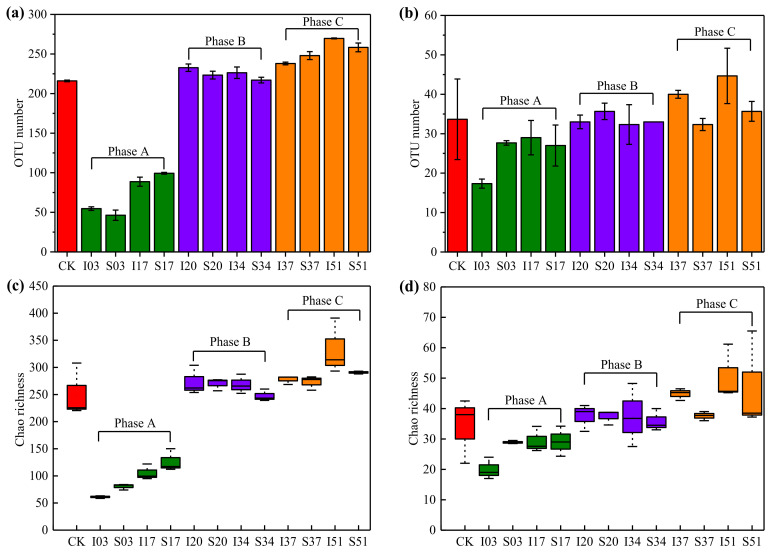
The OTU number of collected samples: (**a**) bacteria and (**b**) fungus, and the Chao richness index of collected samples: (**c**) bacteria and (**d**) fungus. Mean value (n = 3) with standard deviation.

**Figure 7 materials-15-01114-f007:**
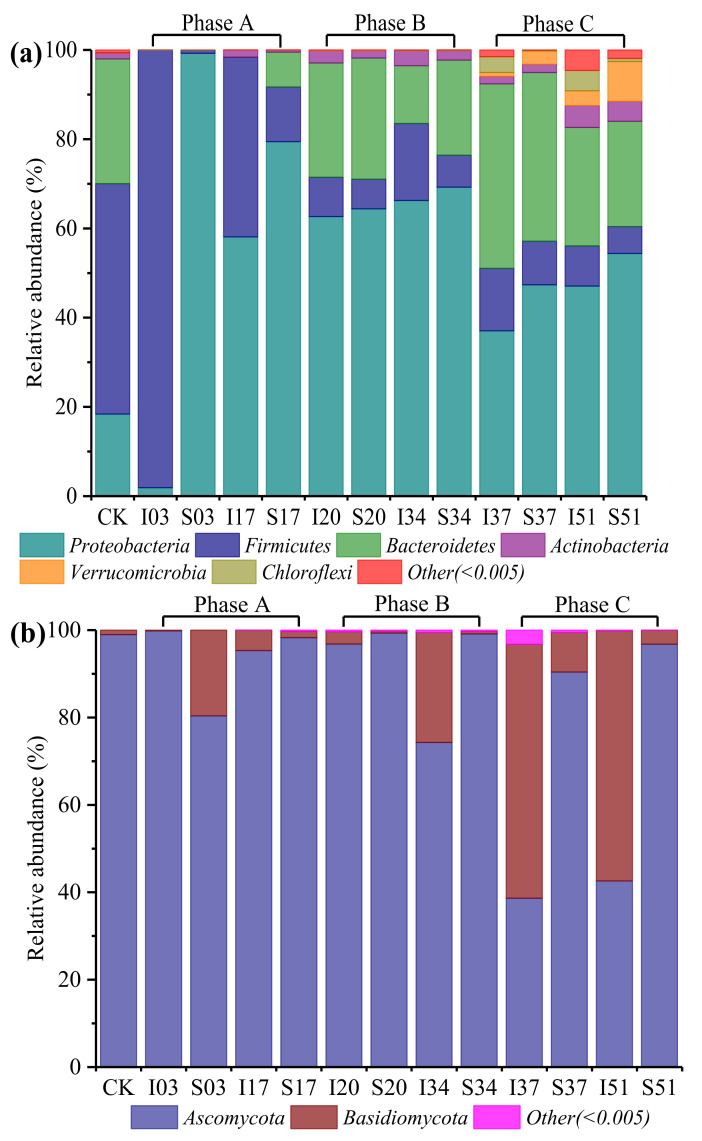
Microbial community composition of (**a**) bacteria phyla, (**b**) fungus phyla.

**Table 1 materials-15-01114-t001:** Experiment process and samples collected in the self-heating experiment.

Phase	Treatment	Days	Sample ID	Temperature (°C)
Raw sawdust (Day 0)	-	0	CK ^a^	30.6
A (Days 1–17)	1. Watered and Turned over (At 0st day after sampling)	03	I03 ^b^	62.0
			S03 ^c^	58.5
		17	I17	29.4
			S17	28.0
B (Days 18–34)	2. Turned over (At 17th day after sampling)	20	I20	33.8
			S20	30.8
		34	I34	30.5
			S34	27.8
C (Days 35–51)	3. Watered and Turned over (At 34th day after sampling)	37	I37	40.7
			S37	33.0
		51	I51	33.6
			S51	29.3

^a^ CK: Check sample; ^b^ I: Inside sample; ^c^ S: Surface sample.

**Table 2 materials-15-01114-t002:** Activation energies of poplar sawdust samples calculated by Friedman method.

Sample	α	*E* (kJ/mol)	Mean *E* (kJ/mol)
Poplar sawdust stack	20%	130.2 (±4.2)	140.5 (±3.7)
	30%	136.8 (±2.2)	
	40%	154.3 (±6.9)	
	50%	140.8 (±1.4)	

**Table 3 materials-15-01114-t003:** The calculated CATs of poplar sawdust samples with different stack dimensions based on FK theory.

**Property**	**Sample**	**Stack Dimensions *l* (m)**					
		0.1 *	0.2	0.4	0.8	1.6	3.2
CAT/°C (±0.01)	Poplar sawdust	175.00	158.27	142.78	128.39	115.12	102.46

* l = 0.1 m was measured from the self-heating substances test experiments.

**Table 4 materials-15-01114-t004:** The component content variations of the inside and surface sawdust samples collected from the compost bin during the self-heating process.

Day	CK	Phase A		Phase B		Phase C	
	0	3	17	20	34	37	51
Inside							
Cellulose (%)	45.92	43.01	41.05	39.87	37.56	36.12	34.32
Hemicellulose (%)	20.89	19.58	18.01	17.64	16.95	16.42	15.45
Lignin (%)	25.44	24.95	24.34	24.08	23.36	23.01	22.26
Others (%) *	7.75	12.46	16.60	18.41	22.13	24.45	27.97
Surface							
Cellulose (%)	45.92	43.65	40.23	38.95	37.31	35.58	33.20
Hemicellulose (%)	20.89	19.72	17.82	17.32	16.45	15.94	14.86
Lignin (%)	25.44	25.01	24.23	23.97	23.12	22.81	21.98
Others (%)	7.75	11.62	17.72	19.76	23.12	25.67	29.96

* Others (%) = 100% − (Cellulose (%) + Hemicellulose (%) + Lignin (%)).

**Table 5 materials-15-01114-t005:** Daily average decomposition rates (%) of cellulose, hemicellulose and lignin of the inside and surface sawdust samples during the self-heating process.

Days	Phase A		Phase B		Phase C	
	1–3	4–17	18–20	21–34	35–37	38–51
Inside						
Cellulose (%)	0.97	0.14	0.39	0.17	0.48	0.13
Hemicellulose (%)	0.44	0.11	0.12	0.05	0.18	0.07
Lignin (%)	0.16	0.04	0.09	0.05	0.12	0.05
Total (%)	1.57	0.29	0.60	0.27	0.78	0.25
Surface						
Cellulose (%)	0.76	0.24	0.43	0.12	0.58	0.17
Hemicellulose (%)	0.39	0.14	0.17	0.06	0.17	0.08
Lignin (%)	0.14	0.06	0.09	0.06	0.10	0.06
Total (%)	1.29	0.44	0.69	0.24	0.85	0.31

## Data Availability

Data available on request.

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
