# Peer review of "Poplar Sawdust Stack Self-Heating Properties and Variations of Internal Microbial Communities"

_materials, 2022, doi:10.3390/ma15031114_

Round 1

Reviewer 1 Report

Interesting topic, broad spectrum of research, and correct methodically performed experiments. My comments:

  1. I propose to modify the title. In my opinion, a shorter but at the same time more specific title will attract more readers "Influence of Poplar Sawdust Stack self-Heating to the changes in its internal microbial community".
  2. I suggest the first sentence in Section 2.2: "In the experiment a laboratory device for self-heating measurement was used (model: HWP01-10E, manufacturer: Hangzhou Young Instrument Science & Technology, Co., Ltd., [city], China), whose oven temperature could be set below 200 ° C”. I have a general comment, I postulate that for all measuring devices used, the following should be stated consistently: (model, manufacturer, city, country).
  3. In Section "2.4. Sawdust stack self-heating experiments" the range of temperature changes and the mean temperature as well as the range of changes in the relative humidity of the ambient air during the 51 days of the experiment should be given (the information in Fig. 4 is not sufficient).
  4. 7. It is unreadable, resolution is too low.
  5. Please consider adding the fourth conclusion that comes to my mind after reading the article: "the described research results are directly applicable in the composting and biodegradation planning of poplar dust and also in the creation of modern biomaterials, which are a combination of chitinous mycelium and lignocellulosic waste, and are used in construction industry, as a packaging material or in furniture".

Sicerelly

Author Response

Response to Reviewer 1 Comments

Point 1: I propose to modify the title. In my opinion, a shorter but at the same time more specific title will attract more readers "Influence of Poplar Sawdust Stack self-Heating to the changes in its internal microbial community".

Response 1: Thank you for your advice! We believe that the title should indeed be shorter. Therefore, after combining your suggestions, we have revised the title to “Poplar Sawdust Stack Self-heating Properties and Variations of Internal Microbial Communities”.

Point 2: I suggest the first sentence in Section 2.2: "In the experiment a laboratory device for self-heating measurement was used (model: HWP01-10E, manufacturer: Hangzhou Young Instrument Science & Technology, Co., Ltd., [city], China), whose oven temperature could be set below 200 °C”. I have a general comment, I postulate that for all measuring devices used, the following should be stated consistently: (model, manufacturer, city, country).

Response 2: Thank you for your advice! We have added relevant information for all measuring devices mentioned in the manuscript by the format (model, manufacturer, city, country).

Point 3: In Section "2.4. Sawdust stack self-heating experiments" the range of temperature changes and the mean temperature as well as the range of changes in the relative humidity of the ambient air during the 51 days of the experiment should be given (the information in Fig. 4 is not sufficient).

Response 3: Thank you for your advice! We have also noticed this problem and agree to supplement the relevant data in section "2.4. Sawdust stack self-heating experiments".

Line 139-142:

The sawdust stack self-heating experiment was done continuously for 51 days. In the 51 days, the ambient temperature varied from 21.5°C to 33.4°C and the mean temperature was 27.0°C, the relative humidity of the ambient air ranged from 25.7% to 84.0% and the average humidity was 61.5%.

Point 4: Fig. 7. It is unreadable, resolution is too low.

Response 4: Thank you for your advice! In order to make the information in Figure 7 more clear, we not only improve the resolution of the image, but also enlarge the image. Meanwhile, Fig. 7 (c) and Fig. 7 (d) were separately placed in the appendix and renamed as Fig. A1. and Fig. A2 respectively.

Point 5: Please consider adding the fourth conclusion that comes to my mind after reading the article: "the described research results are directly applicable in the composting and biodegradation planning of poplar dust and also in the creation of modern biomaterials, which are a combination of chitinous mycelium and lignocellulosic waste, and are used in construction industry, as a packaging material or in furniture".

Response 5: Thank you for your advice! All authors believed after discussion that the conclusion you provided is not suitable as one of the main conclusions of this manuscript. On the other hand, we think your suggestion is valuable and decided to add relevant content in the Introduction part to introduce the application of lignocellulosic waste.

Line 40-42:

These lignocellulosic wastes, including sawdust, straw, bagasse and so on, are valuable for energy utilization and soil nutrient replenishment, and are used as feedstock for biofuels and growth substrates for mycelium-based composites [2, 3].

Reviewer 2 Report

I reviewed article entitled  “The potential risk caused by sawdust stack self-heating with the analysis of physicochemical properties and microbial communities” by Zitao et al. This is quite comprehensive work covering substantial amount of information on the subject discussed. I am sure some of the readers of this journal will be interested in materials presented in the work. I have some minor comments and questions as I listed below, once manuscript is revised based on them, I guess it will be ready for publishing.

Authors should avoid using parentheses within the text, you can blend them seamlessly in the form of  sentence.  There are many within the manuscript several  examples :

Table 1 Each sample (inside and surface, respectively) was……

Component (cellulose……)

Try to use  “namely,” rather  than parentheses.

((as shown in Fig. 7(c))

In Conclusion:

“In short”  use “consequently”

Not TLOEDO  it should be TOLEDO

3.1.2 FK  ?? what is this ? some readers may not know it so you should first fully spell it.

I

((as shown in Fig. 7(c))

In Conclusion:

“In short”  use “consequently”

Author Response

Response to Reviewer 2 Comments

Point 1: Authors should avoid using parentheses within the text, you can blend them seamlessly in the form of sentence. There are many within the manuscript several examples :

(1). Table 1 Each sample (inside and surface, respectively) was……

(2). Component (cellulose……)

(3). ((as shown in Fig. 7(c))

Try to use “namely,” rather than parentheses.

Response 1: Thank you for your advice! We are also aware of this problem and we agree to reduce the use of parentheses in the manuscript. We have revised the manuscript so that most of the information in parentheses is incorporated into sentences. However, in order to keep the content of the manuscript concise and easy to understand, we retain a few parentheses.

Point 2: In Conclusion: ”In short” use “consequently”.

Response 2: Thank you for your advice! We have changed the "In short" in the manuscript into "consequently".

Line 518:

Consequently, during the sawdust stack self-heating process, the sawdust stack temperature between 25℃ and 42℃ was suitable for the growth of most microorganisms.

Point 3: Not “TLOEDO”, it should be “TOLEDO”.

Response 3: Thank you for your advice! We have also noticed this issue and corrected it immediately.

Line 116-118:

In this study, the TG curve of sawdust samples were obtained by the simultaneous thermal analyzer (model: TG/DSC 3+, manufacturer: Mettler TOLEDO, Zurich, Switzerland).

Point 4: 3.1.2 FK ? what is this ? some readers may not know it so you should first fully spell it.

Response 4: Thank you for your advice! We also noted this issue and added its full name in 3.1.2. Meanwhile, we have also added citations to the manuscript to explain the Frank-Kamenetskii theory.

Line 251-253:

3.1.2. CAT of larger sawdust stack derived by Frank-Kamenetskii (FK) theory

       The FK theory can reflect the temperature distribution within the stack of substances [29].

This manuscript is a resubmission of an earlier submission. The following is a list of the peer review reports and author responses from that submission.